# Proteomic and Transcriptomic Responses of the Desiccation-Tolerant Moss *Racomitrium canescens* in the Rapid Rehydration Processes

**DOI:** 10.3390/genes14020390

**Published:** 2023-02-02

**Authors:** Yifang Peng, Tianyi Ma, Xin Wang, Meijuan Zhang, Yingxu Xu, Jie Wei, Wei Sha, Jing Li

**Affiliations:** 1College of Life Sciences, Northeast Agricultural University, Harbin 150030, China; 2The Key Laboratory of Resistance Genetic Engineering and Coldland Biodiversity Conservation, College of Life Sciences, Agriculture and Forestry, Qiqihar University, Qiqihar 161006, China

**Keywords:** *Racomitrium canescens*, desiccation-tolerant bryophyte, rehydration, proteomics, transcriptomics

## Abstract

The moss *Racomitrium canescens* (*R. canescens*) has strong desiccation tolerance. It can remain desiccated for years and yet recover within minutes of rehydration. Understanding the responses and mechanisms underlying this rapid rehydration capacity in bryophytes could identify candidate genes that improve crop drought tolerance. We explored these responses using physiology, proteomics, and transcriptomics. Label-free quantitative proteomics comparing desiccated plants and samples rehydrated for 1 min or 6 h suggesting that damage to chromatin and the cytoskeleton had occurred during desiccation, and pointing to the large-scale degradation of proteins, the production of mannose and xylose, and the degradation of trehalose immediately after rehydration. The assembly and quantification of transcriptomes from *R. canescens* across different stages of rehydration established that desiccation was physiologically stressful for the plants; however, the plants recovered rapidly once rehydrated. According to the transcriptomics data, vacuoles appear to play a crucial role in the early stages of *R. canescens* recovery. Mitochondria and cell reproduction might recover before photosynthesis; most biological functions potentially restarted after ~6 h. Furthermore, we identified novel genes and proteins related to desiccation tolerance in bryophytes. Overall, this study provides new strategies for analyzing desiccation-tolerant bryophytes and identifying candidate genes for improving plant drought tolerance.

## 1. Introduction

Drought is a chronic environmental stress that affects plant development, severely reducing crop yields worldwide [1]. However, many wild plants have evolved to accommodate drought tolerance. This trait has garnered substantial attention within the context of identifying biomolecular patterns underlying its expression, aiming to enhance drought tolerance in crops [2]. During a drought, drought-tolerant plants generally reduce water loss to maintain cell turgor and critical biochemical processes, or enter a dormant state, slowing or stopping their metabolic reactions until the drought ends; they then recover after rehydration. The latter strategy, known as desiccation tolerance, primarily allows plants to endure longer periods without water [3]. Although the mechanisms underlying drought tolerance in plants have been researched extensively, research on desiccation tolerance has been limited [4,5]. Therefore, a better understanding of the genes involved in desiccation tolerance in plants could facilitate new research avenues for improving the drought tolerance of various crops.

Plants with desiccation tolerance are commonly known as “resurrection plants.” Despite this name, it is noteworthy that most vascular resurrection plants require a preparation stage; in vascular plants, this stage entails a slow loss of water until the desiccation tolerance is completely activated throughout the plant’s physiological systems. Therefore, these plants cannot survive rapid water loss [3,6,7]. Unlike vascular plants, resurrection bryophytes are not only capable of surviving rapid water loss, but also rapidly recover once water is available; this occurs through a mechanism that remains largely unknown [3,6,7].

Of all plant groups, bryophytes host the most experimentally validated desiccation-tolerant species, comprising more than 210 species or ~1% of all bryophytes [3]. Of all the known desiccation-tolerant bryophytes, twisted moss (*Tortula ruralis*, also known as *Syntrichia ruralis*) is the most extensively studied. This moss has the greatest known desiccation tolerance, and the mechanisms underlying the tolerance have been investigated at the morphological, physiological, and molecular scales [8]. Overall, the mechanism consists of three stages: constitutive cellular tolerance, repair, and recovery upon rehydration [8].

Bryophytes have a simpler tissue organization than vascular plants. This organizational simplicity establishes the possibility that their desiccation tolerance largely evolved to protect their fragile physical features. Unlike in some desiccation-tolerant angiosperms, in desiccation-tolerant bryophytes, the onset of rehydration is marked by cellular damage [8]. During dehydration, the membranes of desiccation-sensitive plant cells are damaged, whereas those of desiccation-tolerant plants retain their integrity unless dehydration occurs rapidly [9]. However, at the onset of rehydration, the tissues of bryophytes offer only a limited barrier against water flow into cells, which triggers the leakage of cytosol, whereas the more complex tissue organization of desiccation-tolerant angiosperms mitigates this damage [10,11].

The ultra-structures within bryophyte cells may also suffer damage during rehydration, whereas in *T. ruralis*, plastids diminish in size upon dehydration and recover within the first 5 min of rehydration. Thylakoid size does not change upon dehydration; however, thylakoids become inflated, and burst during rehydration, as in other desiccation-tolerant bryophytes [10,12]. The cristae of mitochondria are similarly damaged, but is independent of the dehydration speed [10]. Damage to thylakoids and cristae slows the recovery of energy metabolism after rehydration, and these organellar structures are prioritized for repair within 24 h of rehydration [13].

Within the context of investigating the desiccation tolerance of bryophytes, their cellular repair in response to rehydration has been a challenging research avenue. However, some progress has been made, largely through studying *T. ruralis*. For instance, their anabolism has been shown to recover quickly during rehydration in many bryophytes [6]. Specifically, in the first 2 h of rehydration in *T. ruralis*, protein accumulation profiles change drastically, with the biosynthesis of hydrin proteins being terminated or reduced, whereas that of rehydrins began to increase. Hydrin and rehydrin biosynthesis were once thought to be driven by independent mechanisms [11]. Although hydrins and rehydrins are involved in rehydration, this mechanism has not been extensively studied at a molecular level; this lack of research also encompasses dehydrins, which are late-embryogenesis abundant (LEA) soluble proteins that regulate water conductance [3].

Dehydrins accumulate during the rehydration of *T. ruralis*, possibly to support repair and recovery [14]. During rehydration, dehydrin genes exhibit different expression kinetics depending on dehydration speed, peaking in transcript levels within 1 h of the rehydration in response to rapid dehydration. In contrast, these genes show a delayed peak (i.e., 4–6 h) when the dehydration occurred slowly [15]. To some extent, the rapid response of protein biosynthesis to rehydration in desiccation-tolerant bryophytes is believed to reflect a need to repair the photosynthetic apparatus [16]. How quickly resurrection plants return to full photosynthetic capability is an important factor for evaluating desiccation tolerance; for example, this recovery may be as rapid as 10–20 min into rehydration [17,18].

Establishing the identities of proteins that differentially accumulate during the desiccation and rehydration of *T. ruralis* is important. It is also important to determine how these proteins collectively contribute to desiccation tolerance. The advent of high-throughput “omics” technologies facilitates the exploration of the transcriptome, proteome, and metabolome of most species subjected to any set of environmental conditions, including *T. ruralis* during its desiccation and rehydration. Presently, limited information is available on the proteome dynamics of desiccation-tolerant plants, although there have been some reports on resurrection angiosperms [4,19,20]. However, these reports are not an accurate representation of bryophytes. The transcriptome of desiccation-tolerant bryophytes, specifically *T. ruralis*, has largely been explored [15,21]. In *T. ruralis* gametophytes subjected to rapid desiccation followed by rehydration, candidate genes potentially involved in desiccation tolerance were identified and found to be related to transmembrane activities, phosphorylation, signal transduction, protein biosynthesis, and membrane structure metabolism [21]. A decade later, de novo sequencing of the transcriptome of tortula moss (*S. caninervis*) [22] provided another collection of unigene models from desiccation-tolerant moss, although the moss samples were not exposed to dehydration or rehydration before transcriptome sequencing. Various *S. caninervis* transcripts reported to be related to dehydration and rehydration were later shown to enhance resistance to various stresses in other plants [23,24,25]. The transcriptomic analysis of another desiccation-tolerant bryophyte, silvery bryum (*Bryum argenteum*), established gene expression changes in response to rapid desiccation and rehydration after 2 h or 24 h, thus illuminating both short- and long-term rehydration responses and revealing candidate genes for subsequent analyses [26,27]. Although invaluable, these early transcriptome studies focused only on the later stages of rehydration, leaving a sizable gap in our knowledge of the initial events (within minutes) following the onset of rehydration.

Hoary fringe-moss (*Racomitrium canescens*) is a globally distributed, desiccation-tolerant moss species belonging to the family Grimmiaceae [3,28]. *R. canescens* is resistant to desiccation, possessing a good water storage capacity (i.e., 8–10 times its dry weight) [29,30]; however, the mechanism underlying its desiccation tolerance and rapid rehydration capacity has not been explored in detail. *R. canescens* is the mostly ubiquitous moss species in the Wudalianchi volcanic landforms in Heilongjiang Province, China. This moss sparked our interest because of its remarkable recovery, which we observed in desiccated samples that had been stored for years. Specifically, the dried plants visibly recovered from desiccation within minutes and resumed growth.

In order to discover the morphological, physiological, and molecular responses of *R. canescens* during rehydration, in this study, we used freshly desiccated *R. canescens* plants. This homogeneity in this pioneering study is intended to minimize the effect of confounding variables. These samples were then rehydrated over 6 d. We also performed physiological and biochemical assays on the harvested samples, establishing the specific physiological stages of rehydration. We then explored the proteome of the desiccated *R. canescens* plants, and as well as plants within 1 min of the process of rehydration, to identify proteins that differentially accumulated in response to rehydration. Lastly, we performed independent transcriptome analyses representing different time points in the rehydration of *R. canescens* (at 1 min, 30 min, 45 min, 6 h, and 1 d). Because the *R. canescens* genome is not available, we performed a de novo assembly of all RNA-seq reads to define 94,691 unigenes, of which we functionally annotated 57,367 (60.6%). Placing all proteomic and transcriptomic results into context, we predict that different structures and functions of *R. canescens* cells recover under different kinetics. This study will empower a renewed exploration of desiccation tolerance in resurrection bryophytes and provide essential information on genes and proteins that contribute to this process.

## 2. Materials and Methods

### 2.1. Plant Materials and Desiccation Treatment

The *R. canescens* samples and their underlying substrata were collected from the Laoheishan region of Wudalianchi National Scenic Area (126°00′–126°25′ E, 48°30′–48°50′ N) in Heilongjiang Province, China, in May 2016. Once brought to the laboratory, the plants were laid on a clean tile floor surface and desiccated by air-drying for 14 d. The plants were then stored at ~25 °C and 40–50% air humidity in the dark. Before desiccation, the stored plants were first rehydrated and cleaned in sterile deionized water in plastic trays in a plant culture room at 25 °C ± 1 °C, 50–60% humidity, and a photoperiod of 12 h light/12 h dark. After growth for 7 d, plants were evenly split in separate sets of 6 g (fresh weight). For the rapid desiccation treatment, a drying apparatus (Supin Laboratory Instruments Ltd., Nantong, China) with a diameter of 40 cm and a depth of 30 cm was used, with allochroic silica gel beads as desiccant. The plants were then gently immersed into silica beads. Finally, the plants were completely desiccated for 24 h.

### 2.2. Rehydration Treatments and Morphological Observations

Desiccated *R. canescens* plants were taken out of silica gel for rehydration treatments and immediately transferred to glass Petri dishes (20 cm diameter) containing 100 mL of sterile deionized water. The plants were then placed in a culture room at 25 ± 1 °C, 50–60% humidity, and a photoperiod of 12 h light/12 h dark. The plants were photographed with a digital camera (EOS 200D, Canon, Japan) just before rehydration and after initiation of rehydration at 1 min, 3 min, 6 min, 15 min, 30 min, 45 min, 1 h, 3 h, 6 h, 12 h, 1 d, 3 d, and 6 d; the samples were also collected at each time point for further analysis and labeled as R0min, R1min, R3min, R6min, R15min, R30min, R45min, R1h, R3h, R6h, R12h, R1d, R3d, and R6d, respectively. Water content and relative water content (RWC) were determined for whole plants. For other analyses, the green tender parts containing stems with leaves of *R. canescens* gametophytes were excised from the base, frozen in liquid nitrogen, and stored at –80 °C.

### 2.3. Physiological and Biochemical Analyses

The samples were weighed immediately after removal from the Petri dishes. The RWC; contents of free proline, soluble sugars, soluble proteins, and malonaldehyde (MDA); and the activities of catalase (CAT), peroxidase (POD), and superoxide dismutase (SOD) were determined as described by Yin et al. [31].

Briefly, the RWC was calculated as follows: RWC = [(fresh weight (FW) − dry weight (DW)]/[saturated wight (SW) − DW]. The FWs of the materials were measured immediately after treatments; SW was measured after the materials achieved saturation by immersion in deionized water for 2 d; DW was recorded after the materials were oven-dried at 90 °C for 30 min and then vacuum-dried at 60 °C until the weight was constant [31].

For proline assay, the samples were ground in 5 mL of 3%sulfosalicylic acid, then incubated for 60 min at 100 °C. After cooling to room temperature (RT), homogenate was centrifuged for 5 min at 15,000 r/min in order to collect the supernatant. Then, 2 mL of glacial acetic acid and 2 mL of ninhydrin were added into 1 mL of supernatant; after boiling for 30 min and being cooled down to RT, 4 mL of methylbenzene was added. After being incubated for 2 h, the upper layer solution was collected, and the absorbance reading at 520 nm was measured using a spectrophotometer. Proline standard curves were used for the concentration determination; the proline content was calculated as follows: proline content (mg/g DW) = concentration (C) × volume of total extraction solution (V)/DW [31].

For soluble sugar assays, the samples were grounded in 5 mL deionized water, then incubated 30 min at 100 °C. After cooling down to RT and being centrifuged for 5 min at 15,000 r/min, the supernatant was collected and diluted to 50 mL using deionized water. Then, 0.2 mL of diluted extraction solution was mixed with 5 mL of concentrated sulfuric acid and 0.5 mL of 2% (w/v) ethyl acetate solution of anthrone. The mixture was incubated at 100 °C for 1 min; after cooling down to RT, the absorbance at 630 nm was measured using a spectrophotometer. Glucose standard curves were used to determine the soluble sugar concentration; the total soluble sugar content was calculated as follows: soluble sugar content (μmol/g DW) = C × V/DW [31].

The samples for extracting MDA were incubated in the solution containing 0.6% thiobarbituric acid and 10% trichloroacetic acid at 100 °C for 5 min. The absorbance was measured at 450 nm, 532 nm, and 600 nm of the supernatant, and recorded as OD_450_, OD_532_, and OD_600_, respectively. The MDA concentrations were calculated as follows: MDA concentration (μmol/g DW) = [6.45 × (OD_532_ − OD_600_) − 0.56 × OD_450_] × V/DW [31].

The total soluble protein content of samples and protein content of enzymatic preparations were extracted using phosphate-buffered solution (pH 7.8) and determined using the Bradford method, as described previously [31,32]. For protein extraction, the samples were ground with the solution of 50 mM of phosphate-buffered solution (pH 7.8) with 2% polyvinylpyrrolidone at 4 °C, then centrifuged at 20,000× *g* at 4 °C for 15 min. CAT activity was determined by measuring H_2_O_2_ consumption at 240 nm, POD activity was determined using a guaiacol mothed at 470 nm, and SOD activity was determined by inhibiting the photochemical reduction of nitroblue tetrazolium at 560 nm, as previously described [31].

Water content was calculated using (FW–DW)/FW. For all biochemical assays, 0.2 g of frozen *R. canescens* material was used for each assay, and at least three biological replicates were performed. Relative changes are shown as percentages to allow the reader to better visualize fold changes and trends between time points. All values were normalized to dry weight.

### 2.4. Label-Free Quantitative Proteomic Analysis

Frozen R0min, R1min, and R6h samples (three independent biological replicates each) were ground to a powder in liquid nitrogen. The total proteins were then extracted from 6 g of fresh tissue per sample following the cold acetone method as previously described [33]. The proteins were then digested with sequencing-grade modified trypsin (Promega, Madison, WI, USA) overnight, following the manufacturer’s protocol. The digested peptides were separated by nano-liquid chromatography and analyzed through on-line electrospray tandem mass spectrometry (ESI-MS/MS), as described previously [34]. The mass spectrometer was operated in the data-dependent acquisition mode to automatically switch between MS and MS/MS acquisition. The full-scan MS spectra (m/z 350–1550) were acquired with a mass resolution of 35 D, followed by sequential high-energy collisional dissociation MS/MS scans with a resolution of 17.5 D. The dynamic exclusion time was set as 20 s.

After the MS/MS spectra were extracted, a charge-state deconvolution and deisotoping step was performed, and the data were then transformed into MGF files using Proteome Discovery 1.2 (Thermo Fisher, Pittsburgh, PA, USA) and processed by MaxQuant v1.5, as described previously [34]. PEAKS DB was configured to search the protein sequences in the NCBI non-redundant (NR) protein database (http://www.ncbi.nlm.nih.gov, accessed on 20 May 2020), Swiss-Prot protein database (http://www.expasy.ch/sprot, accessed on 20 May 2020), Kyoto Encyclopedia of Genes and Genomes (KEGG) database (http://www.genome.jp/kegg, accessed on 20 May 2020), and KOG database (http://www.ncbi.nlm.nih.gov/COG, accessed on 20 May 2020), which were then annotated again with the transcriptome data. The peptides were filtered using a 1% false discovery rate (FDR). Protein identification was accepted if at least two identified peptides were present. Proteins containing similar peptides that could not be differentiated based on MS/MS analysis alone were grouped to satisfy the principles of parsimony.

The mass spectrometry proteomics data were deposited in the ProteomeXchange Consortium via the PRIDE [35] (http://www.ebi.ac.uk/pride, accessed on 9 January 2023) partner repository with the dataset identifier PXD026286. The intensity-based absolute quantification (iBAQ) values [36] were used for the calculation of peptide and protein abundance calculations. Proteins detected in all three replicates or in none of the three replicates were used for comparison. Proteins with a fold change of >1.2 or <0.83 and significance level of *p* < 0.05 were considered to be differentially accumulating.

### 2.5. Transcriptome Deep Sequencing (RNA-Seq) Analysis 

The total RNA was extracted from 1 g of frozen tissue, collected as three independent biological replicates, for samples R0min, R1min, R30min, R45min, R6h, and R1d; the RNeasy Plant Mini Kit (QIAGEN, Hilden, Germany) was used following the manufacturer’s instructions. The concentration and quality of total RNA were determined using a DS-11+ fluorometer (DeNovix Inc., Wilmington, DE, USA) and electrophoresis (1% agarose gel). The pretreatment of the mRNAs, cDNA synthesis, and sequencing program were performed as previously described [34], and cDNA libraries were sequenced on an Illumina HiSeq 4000 system (Illumina Inc., San Diego, CA, USA) by Gene Denovo Biotechnology Co. (Guangzhou, China).

Raw sequencing treads were trimmed by fastp [37]; adapters and reads with over 10% unknown nucleotides or with a low-quality score for 40% or more nucleotides (*Q*-value ≤ 10) were removed. Clean, high-quality reads were used for the de novo assembly of unigenes using the TRINITY software package, as described previously [38]. The assembled unigenes were annotated using the Basic Local Alignment Search Tool (BLASTx) (http://www.ncbi.nlm.nih.gov/BLAST/, accessed on 20 May 2020) against the NR, Swiss-Prot, KEGG, and KOG databases. The dataset is available from the NCBI Short Read Archive (http://www.ncbi.nlm.nlm.nih.gow/sra, accessed on 9 January 2023), and its accession number is SRP319802.

The unigenes were expressed in reads per kilobase of transcript per million mapped reads (RPKM), as described previously [39]. Differentially expressed genes (DEGs) were defined as those with a fold change of >2 between two samples and an FDR of <0.05, as determined using the edgeR package [40]. Only the unigenes detected in all three replicates or none of the three replicates were used for comparison. Upregulated and downregulated DEGs between groups were subjected to Gene Ontology (GO) and KEGG pathway analyses, as described previously [41]. For trend analysis, the data from each sample were normalized relative to the R0min sample, parameters were set (-pro 20 -ratio 1.0 [log_2_(2) = 1, log2(1.5) = 0.5849625, log2(1.2) = 0.2630344]), and samples were clustered into 26 profiles using the Short Time-Series Expression Miner software, as described previously [42]. Unigenes with significant (*p* < 0.05) expression profiles were subjected to GO and KEGG pathway enrichment analysis.

### 2.6. Quantitative Real-Time PCR Analysis

Twelve DEGs were randomly selected from significant expression profiles for quantitative real-time PCR (qPCR) to confirm the accuracy of the RNA-seq quantification. The primer sequences used for qPCR are listed in Appendix A. Unigene0044326, which encodes an alpha-tubulin chain, was used as the internal control. A total of 10 μL of TB Green Premix Ex Taq II (Tli RNaseH Plus) (Takara Biomedical Technology, Beijing, China) reagent was added to each reaction according to the manufacturer’s instructions, and the reactions were performed and analyzed using CFX90 (Bio-Rad, Hercules, CA, USA), as previously described [43,44]; the relative expression of each gene in the R24h sample was set as 1. The Pearson correlation coefficient was calculated to determine the correlation between the qPCR and RNA-seq data using IBM Statistical Package for the Social Sciences (SPSS) software (version 20.0; IBM, New York, NY, USA). Graphs were drawn using Origin 2019b software (OriginLab, Northampton, MA, USA).

## 3. Results

### 3.1. Physiological and Biochemical Changes during the Rehydration of Desiccated R. canescens

*R. canescens* plants showed remarkable recovery from desiccation, as evidenced by their morphology, with leaves unfolding and stems pointing up within 1 min. Plant size increased sharply within the first 3 min, and then slowly thereafter (Figure 1). *R. canescens* plants rehydrated for 1 d were noticeably larger than plants rehydrated for only 12 h (Figure 1). This was indicative of a return to normal growth, rather than being driven simply by water intake, which is consistent with the sharp, rapid rise in water content in rehydrated plants (see below: Figure 2A,B).

The desiccated *R. canescens* initially had a water content of only 5–6%, which rose sharply to over 70% within 1 min of rehydration. The water content then remained at ~70–80% for 6 d (Figure 2A). Thus, desiccated *R. canescens* plants can rapidly absorb water. Similarly, the RWC of desiccated *R. canescens* plants was <10%, but it increased rapidly to >70% as early as 1 min and remained at this level until 1 d into rehydration (Figure 2B). RWC only dropped significantly 3–6 d into the rehydration regime (Figure 2B), although water content remained unchanged (Figure 2A). We hypothesize that the initial rise in water content and RWC reflects the rapid rehydration kinetics mentioned above, while the later reduction in RWC signals the resumption of plant growth.

We also determined the cellular contents of the osmoprotectants proline, soluble sugars, and soluble proteins (Figure 2C–E). The proline content increased almost 5-fold over the first 6 min and then decreased slightly between 15 min and 30 min after initiation of rehydration, only to rise again after 45 min. Within 1 h of the onset of rehydration, proline content stabilized at ~260% of the levels observed in desiccated plants, and then decreased gradually after 1 h to 6 d from the commencement of rehydration (Figure 2C). The accumulation profile of soluble sugars was much simpler: their levels began to rise 3 min after onset of rehydration and remained fairly constant, at ~150% of those seen in desiccated plants (Figure 2D). Similarly, after 1 min of rehydration, soluble protein content reached 150% of the levels seen in desiccated plants; the proteins increased over 6 h, reaching 200% of desiccated plant levels, then decreasing moderately and remaining relatively stable thereafter (Figure 2E).

The levels of MDA, a marker of oxidative stress and thus membrane integrity, dropped by ~70% after only 1 min of rehydration relative to that of desiccated plants, then from 3 min to 1 h increased dramatically to 450% of desiccated plant levels. The MDA content then fluctuated for the remainder of the assessment periods, with peaks at 12 h and 3 d and a trough (of ~300% of desiccated plant levels) at 1 d (Figure 2F).

### 3.2. Proteomics Analysis of R. canescens Plants during the Early Stages of Rehydration

By 6 h into the rehydration, most physiological parameters had reached their maximum levels or begun to stabilize, and plants exhibited close to complete growth recovery. After MS detection, we identified 20,402 peptides, including 17,069 unique peptides, across all plant samples; our results revealed 3433 predicted proteins. The numbers of peptides and proteins detected in each replicate are listed in Appendix A. We quantified each protein using the iBAQ method, and compared their abundance between the R0min, R1min, and R6h samples to identify differentially accumulating proteins (DAPs). A summary of the DAP numbers for each pairwise comparison is shown in Figure 3.

Surprisingly, most proteins detected in the R0min samples underwent a reduction in abundance upon rehydration (Figure 3A), in apparent contradiction to the rise in total soluble protein content over the course of rehydration (Figure 2E). The number of DAPs with higher abundance either in R1min or R6h samples relative to the R0min samples was low, facilitating a direct exploration of their biological functions through annotation. Among the DAPs that were most abundant in rehydrating/rehydrated plants, we detected eight proteases or peptidases (Figure 4A, Appendix A). Several others of these enzymes were highly abundant in the R1min and R6h samples, constituting a large portion of all DAPs with a higher abundance. We also noticed that their abundance either remained high or increased further 6 h into rehydration (Figure 4, Appendix A). As it is difficult to establish whether the amount of decreased DAPs was caused by gene expression regulation or the potential global degradation, we only used proteins whose abundance was high both in in rehydrating and rehydrated plant samples for further functional prediction. These proteins may result in a better understanding of the physiological changes associated with rehydration. A summary of these DAPs is provided in Appendix A, and their relative amounts, GO, and KEGG pathway enrichments analysis are provided in Figure 4. DAPs that were more abundant at both 1 min and 6 h after the onset of rehydration but did not reach the level of significance (*p* < 0.05) in the R1min samples or show a fold change of ≤1.2 relative to the R0min control are also included.

For the reason that most proteins were not annotated with any GO terms or in the KEGG database, most of the enriched pathways or GO terms contain only very few DAPs (Figure 4B–E). Therefore, even when enriched with significance, this was still insufficient to tell the importance of many pathways and GO terms; the effects of independent DAPs will be further analyzed according to other annotated information (Figure 4A, Appendix A). However, from the pathway and GO enrichment results, redox regulation and the degradation of different saccharides were suggested, and happened in the rehydration process to some extent (Figure 4A–C).

### 3.3. Transcriptomics Analysis of R. canescens during Rehydration

To complement our proteomic analysis of the changes associated with rehydration, we conducted an RNA-seq of desiccated *R. canescens* plants (R0min), as well as the plants 1 min (R1min), 30 min (R30min), 45 min (R45min), 6 h (R6h), and 1 d (R1d) into the rehydration process. We selected R1d as a representative of complete rehydration, as this was the last time point before the RWC significantly decreased (Figure 2B). We documented the largest physiological and biochemical changes 30 min after the onset of rehydration relative to other time points (Figure 2), whereas many physiological parameters differed between the 45 min and 30 min time points, providing a rationale for assessing samples from both time points using RNA-seq.

The de novo sequencing result of all *R. canescens* samples with annotations (Appendix A) was used to define unigenes in preparation for expression analysis. We then mapped the clean reads from each RNA-seq sample to the unigenes defined above; the mapping ratio of all the samples was from ~86.74% to ~90.79%. A summary of the number of DEGs for all pairwise comparisons is shown in Figure 5A. For a comprehensive analysis of the physiological responses and metabolic or signaling pathways at different stages of rehydration, we subjected each set of DEGs from pairwise comparisons to GO terms and KEGG pathway enrichment analysis (Appendix A). We also conducted a trend analysis over the entire course of rehydration to identify the co-expressed genes and regulatory modules. We considered unigenes that were differentially expressed in at least three pairwise comparisons for this analysis, resulting in 26 co-expression clusters with a *Q*-value ≤ 0.05, as determined by the Short Time-Series Expression Miner software (Figure 5B,C). Of these 26 clusters, 7 were deemed significant (clusters 0, 8, 13, 16, 20, 23, and 25). We repeated the GO term and KEGG pathway enrichment analyses on their constituent unigenes to better understand their associated biological functions (Appendix A). To confirm the accuracy of transcriptomic quantification, qPCR was performed on 12 DEGs randomly selected from the significant clusters. The qPCR and RNA-seq quantification results mostly correlated well (Figure 6), indicative of the biological validity of the transcriptomic data being highly effective.

As shown in Appendix A, information provided by the GO term and KEGG pathway enrichments was very abundant with complexity. In order to acquire a relatively clear understanding about part of the main responses in the rapid rehydration processes, we firstly further analyzed the expression patterns of the gene categories that drew our attention according to the GO term and KEGG pathway enrichment results (Appendix A). For some desiccation or drought stress-related proteins, including rehydrin, LEA proteins (including dehydrins), early light-inducible proteins (ELIPs), and glutamine synthases (GSs), the levels of the corresponding transcripts were high, and in some cases, had their highest relative levels in the R0min samples (Appendix A). Similar gene expression patterns were also found for some unigenes encoding proteins related to the biosynthesis of γ-aminobutyric acid (GABA) (Appendix A) and brassinosteroids (Appendix A), and abscisic acid (ABA) precursor, xanthoxin (data shown in Appendix A). In addition, many unigenes belonging to KEGG pathways related to the energy metabolism showed high levels of expression in the R0min samples, including those involved in carbon fixation in photosynthetic organisms (ko00710) (Appendix A), oxidative phosphorylation (ko00190), glycolysis/gluconeogenesis (ko00010), and the tricarboxylic acid cycle (TCA) pathway (ko00020) (Appendix A). Some genes involved in trehalose and raffinose biosynthesis showed similar expression patterns (Appendix A).

## 4. Discussion

### 4.1. R. canescens Is a Desiccation-Tolerant Bryophyte That Recovers from Rehydration through Rapid Physiological Changes

Resurrection plants with desiccation tolerance can endure RWC of ≤10% and still recover once rehydrated. Here, we confirmed longstanding reports of desiccation tolerance in the bryophyte *R. canescens* [3] to verify its capacity for efficient rehydration. Specifically, we found that after rapid desiccation, the desiccated materials had an RWC of ~10% or lower, but this quickly increased to ~70% after 1 min of rehydration (Figure 2B), and the physiological functions of the plants were restored thereafter (Figure 1).

Given the rapidity of this recovery, we speculated that it could not be solely controlled by gradual changes in physiological parameters and gene expression. We obtained a preliminary confirmation of our speculation by studying the physiological and biochemical parameters of the desiccated *R. canescens* plants, focusing on the commencement of rehydration (Figure 2). As water absorption seemed the most likely cause of this rapid rehydration, we tested the plants’ water content and RWC. The water content of the samples remained 70–80% over the 6 d (Figure 2A), but the RWC fell to <60% on the third day after rehydration (Figure 2B). We believe that this reflected the resumption of plant growth, culminating in more water being required by the plant.

We then analyzed the osmoregulatory substances influencing water absorption during rehydration, specifically, proline, soluble sugars, and soluble proteins. These substances are all known to accumulate during desiccation in vascular resurrection plants and bryophytes [13,45,46]. Among these, the trajectory exhibited by soluble proteins (Figure 2E) was reminiscent of dehydrin accumulation in *T. ruralis* [8]. Additionally, the changes in other osmoregulatory substances had some aspects in common with those in other desiccation-tolerant bryophytes, including an abrupt pattern of physiological change, which was paralleled by changes in enzyme activity (Figure 2C,D,G–I) [8].

We also assessed the changes in the content of MDA. MDA is the final product in membrane lipid peroxidation; therefore, it indirectly reflects the integrity of the membrane and the organelles in houses. Within the context of our study, we expected that MDA abundance would first reflect damage to membranes that is expected in the early stages of rehydration, and after (i.e., through its reduction) point repair [10,12]. However, the results of this assay were unclear. MDA content sharply increased after 3 min of rehydration, and remained high thereafter, fluctuating between 300% and ~650% (Figure 2F). Alternatively, the MDA levels 1 d after rehydration may represent the natural level, while the increase starting from 15 min after rehydration corresponds to the stressed condition, but the sharp increase after 3 min of rehydration remains difficult to explain (Figure 2F). We speculate that either the MDA was scavenged at that time (i.e., before the sharp increase 3 min into rehydration) or water entered the cells, resulting in their rapid expansion; MDA might have thus been flushed out of the cell because of its hydrophilicity and small size. Since we can only speculate, this phenomenon warrants further research to clarify the effects of the dehydration and rehydration on membrane ultrastructure over time.

### 4.2. Proteomics Analysis Indicates That the Protein Content of R. Canescens Undergoes Substantial Changes during Early Rehydration

Having confirmed the rapidity of the physiological and morphological changes that drive the rehydration of *R. canescens* (Figure 1 and Figure 2), we then explored their associated early recovery responses (i.e., 1 min after rehydration), at which point, plant morphology had already mostly recovered (Figure 1). Protein accumulation was thought to be a crucial aspect of early rehydration in desiccated *T. ruralis*, but was thought to involve a mechanism other than transcriptional regulation [11]. Moreover, we observed that the soluble protein content significantly increased 1 min after rehydration (Figure 2E); therefore, we used proteomics to explore mechanisms underlying recovery.

Our initial quantification of protein levels to identify DAPs revealed that, contrary to our expectations, most DAPs showed a significantly decreased abundance after rehydration (Figure 3A). Upon further analysis, we determined that the most significantly upregulated DAPs, both in number and protein abundance, were proteases and peptidases (Figure 4, Appendix A), which is similar to findings in a proteomic study of *B. argenteum* [19]. Thus, it is unclear whether the decrease in the abundance of a particular downregulated DAP is the result of response regulation or universal protein degradation. We suspect that this is more likely to reflect a general reprogramming of protein expression, as many ribosomal components and functionally related proteins (most of them not detected in the desiccated samples) also significantly increased in the R1min samples (Figure 4, Appendix A), and that this includes universal protein degradation to provide raw material for the synthesis of new proteins. In this case, assessing the putative effects of specific downregulated DAPs might be largely incomprehensive, whereas an investigation of the upregulated DAPs would be much more indicative of the mechanisms underlying rehydration.

The proteins related to the physiological indices assessed here showed generally concordant trends (Appendix A). Many enzymes responsible for saccharide degradation were significantly increased in R1min samples, with the most abundant enzymes being mannosidases, followed by beta-fructofuranosidase, two xylosidases, and trehalase (Appendix A). Mannose and trehalose have been reported to play a role in resurrection of plant desiccation tolerance [47]. Trehalose may be formed and stored in desiccated *R. canescens* and degraded into glucose after rehydration, with mannose and xylose being primarily involved in this process. The mannosidases identified were alpha-mannosidases and a putative mannosylglycoprotein endo-beta-mannosidase (Appendix A), suggesting that mannose is primarily released from mannosylglycoproteins [48].

Two PODs and an SOD were significantly increased in R1min samples, along with changes in sugar content (Figure 2D) and protein activity and abundance (Figure 2E,H,I) in R6h samples (Figure 4, Appendix A). Seeds and pollen grains also exhibit desiccation tolerance, in which rehydration likewise damages the cytoskeleton, the nucleus, and chromatin [49,50]. However, similar observations have not been reported in bryophytes before. In our results, proteins related to cytoskeletal, nuclear, and chromatin structures increased after rehydration (Figure 4A,E, Appendix A), given that this damage may occur in *R. canescens*. Some of these proteins were undetectable in the R0min samples, implying that damage might occur during desiccation, but not after rehydration. Some putative stress-related proteins were significantly upregulated in R1min samples (Figure 4A, Appendix A), suggesting that the cells of R1min samples might have been stressed. We originally intended to investigate membrane-related proteins, including structural components and functional proteins that seemed likely to function in the rehydration process, but our proteomics results did not provide sufficient detail to address these questions, which is why further research is required.

### 4.3. Energy Substances and Classic Desiccation-Tolerant Regulators Accumulated during the Rapid Desiccation of R. canescens

We summarized the probable events by combining the physiological, transcriptomic, and proteomic results (Figure 7). Notably, ~40% of the transcripts could not be annotated using the present databases, and this is a common phenomenon in bryophyte transcriptomics studies [21,22,26,27]. This is primarily because of limited research on the genetics and proteomics of bryophytes. Therefore, the genes and proteins without clear annotation will not be discussed in this study, although many of them performed robust expression changes, warranting their further study in the future.

Soluble sugars and LEA proteins, known as dehydrins, accumulate during dehydration in desiccation-tolerant angiosperms; therefore, the contents of these molecules can be assayed as drought indicators in these species [3]. However, soluble sugars and LEA proteins do not change in response to dehydration in bryophytes [51,52]. These findings suggest that desiccation-tolerant bryophytes employ a constitutive dehydration tolerance mechanism rather than an induced process, implying that these species are prepared for the onset of dehydration at any time, allowing an immediate response.

ELIPs have been reported to accumulate during desiccation stress responses in some resurrection plants, regulating protection from photooxidative damage, and binding and deactivating chlorophyll [53]. GSs act during plant drought stress to regulate the detoxification of NH_4_^+^ [54]. GABA has been reported to regulate drought and desiccation stress and plant growth as a signal molecule in some resurrection plants [55]. Brassinosteroids are important steroid phytohormones that regulate growth, development, and multiple stress responses in angiosperms [56,57]. Additionally, the KEGG pathway “biosynthesis of brassinosteroids” (ko00905) was significantly enriched for many genes we investigated in this study (Appendix A). Therefore, although little is known about the functions of brassinosteroids in bryophytes, our data suggest that brassinosteroids may function in the desiccation tolerance of *R. canescens*. The effects of ABA, another major phytohormone involved in regulating plant stress responses, on bryophyte desiccation stress responses are not clear enough [8,52]; however, as xanthoxin has been reported to function similarly to ABA in some regards [58], we speculate that xanthoxin may play roles in *R. canescens* that are similar to those of ABA in angiosperms.

The expression patterns of unigenes belonging to the KEGG pathways related to the energy metabolism suggested that compounds involved in energy metabolism tended to accumulate during desiccation in *R. canescens* (Appendix A). The transcriptomic results implied that trehalose and raffinose may have been synthesized in desiccated *R. canescens* (Appendix A), which is consistent with previous reports that trehalose and the raffinose family oligosaccharides are involved in the cellular protection of some resurrection plants [59,60].

### 4.4. Vacuolar Components May Play Important Roles in the Initial Stage of Rehydration in R. canescens

Because *R. canescens* rapidly recovers its shape within only minutes of the start of rehydration, and water enters the plants very quickly (Figure 2A,B), we initially suspected that aquaporins might mediate water absorption during rehydration. However, the proteomics results could not provide much guidance in this regard, as aquaporins are membrane proteins that are difficult to extract using most standard protein extraction methods. In the transcriptomic data, 80 unigenes were annotated as encoding aquaporins (Appendix A), 16 of which were significantly upregulated and showed the highest expression in R1min samples (Appendix A). This suggested that they may play a major role in water absorption at the beginning of the rehydration process. Among the 16 unigenes, 4 were annotated as encoding plasma membrane intrinsic proteins (PIPs), and 5 as encoding tonoplast membrane intrinsic proteins (TIPs), representing functional aquaporins located primarily in the plasma membrane and tonoplast, respectively [61]. Moreover, only one PIP and three TIPs showed the highest expression in R1min samples (Appendix A), suggesting that the production of vacuolar aquaporins triggered the entry of water deeper into the cell early during rehydration. Considering that other proteins in vacuolar membranes might also play crucial roles at this time point, we extended our analysis to other DEGs annotated with the GO term vacuolar membrane (GO:0005774) (Appendix A).

The results made us consider whether the expressions of V-type ATPases (V-ATPases) were significantly upregulated in R1min samples, and an analysis of the expression of unigene-coding components of V-ATPases confirmed that this was the case (Appendix A). These results imply that the alteration of vacuolar function may be one of the responses involved in the desiccation tolerance of *R. canescens* (Figure 7C).

### 4.5. Different Functions and Cellular Structures Showed Distinct Patterns of Cell Function Recovery at Different Stages of Rehydration

Before the further analysis of the transcriptomics data, we examined the upregulated DAPs to determine if any increased according to the corresponding DEGs. However, only three DEGs (unigene0007756, unigene0054408, and unigene0064711) showed similar changes in R1h to R0min samples, and five DEGs (unigene0061910, unigene0049397, unigene0037797, unigene0008118, and unigene0041109) in the R6h to R1min samples. Additionally, almost no conformity in significantly enriched KEGG pathways or level 2 GO terms was observed between the proteomic and transcriptomic analyses. Due to the scarce correlation between the transcriptomic and proteomic data, we primarily focused on the transcriptomic and proteomic data individually.

In the R1min samples, the activation of protein synthesis during the rehydration process was further confirmed by the expression of unigenes encoding ribosomal components and proteins related to protein processing in the endoplasmic reticulum (ER) (Appendix A). This is in accordance with the physiological and proteomic results (Figure 2E, Appendix A). Many LEA protein- and ELIP-coding genes were also highly expressed in the R1min samples (Appendix A), as were the unigenes related to the synthesis of xanthoxin, GS, and GABA (Appendix A), in accordance with the stressed conditions early in the rehydration process. In contrast, most unigenes related to photosynthetic reactions were almost completely silenced at this stage (Appendix A), indicating that chloroplast function had not yet recovered. The expression analysis of unigenes related to saccharide biosynthesis and metabolism, sorbitol, mannitol, sucrose-6-phosphate, and trehalose indicated that the corresponding proteins probably accumulated in the R1min samples (Appendix A), which is expected, given that saccharides and their derivatives can help regulate cellular osmotic potential, for instance, through trehalose accumulation in the vacuole [46], as well as being primary sources of glucose for energy (Appendix A). Another type of metabolite predicted to accumulate in R1min samples based on the transcriptomics results was steroids (Appendix A), which are substrates of metabolic pathways that produce many important secondary metabolites, including brassinosteroids and components of membrane lipids [62].

The data from the *R. canescens* samples from later than R1min of rehydration suggest that different functions and cellular structures might be recovered at different stages, possibly through distinct mechanisms. In summary, cell division might be reinitiated, and mitochondrial components and functions are likely to recover in R30min samples (Figure 7D), whereas photosynthesis and plastid components were more likely to recover in R45min samples (Figure 7E), and most other biological functions were likely to be resumed at 6 h after rehydration (Figure 7F).

Based on the soluble protein results, we checked the expression of unigenes predicted to encode ribosomal components; however, the expressions of a few unigenes increased significantly in the R30min samples, compared to those in the R1min samples (Appendix A), indicating that sufficient ribosomes may have been synthesized before this time point. However, most unigenes encoding mitochondrial ribosome components increased at this time, similar to the levels in R6h and R1d samples, but slightly higher than those in R0min, R1min, and R45min (Appendix A).

The unigenes encoding gibberellin 2β-dioxygenases, enzymes responsible for producing gibberellins (GAs) (Appendix A), showed the highest expression in the R30min samples. GAs and the precursor *ent*-kaurene activate cell division, spore germination, and protonema differentiation in the bryophyte [63,64]. These results suggest that the growth of *R. canescens* started to resume around this point. In contrast, most of the unigenes encoding enzymes that directly synthesize indoleacetic acid (IAA) were not expressed at high levels, but many unigenes encoding tryptophan synthases showed similar expression patterns to the mitochondrion-related unigenes mentioned above, which are responsible for the biosynthesis of tryptophan (i.e., a precursor of IAA) (Appendix A).

The morphology was similar between the R45min and R30min samples (Figure 1), but there were significant differences in physiological indices (Figure 2) and gene expression, so we predicted that the biological activities of *R. canescens* would change, even over such a short time. From the transcriptomic results, we propose that the biosynthesis of mannose, sucrose, and trehalose-6-phosphate would be restarted, and trehalose might be used to produce glucose to supply energy (Appendix A). The examination of the unigenes related to the photosynthesis and chloroplast structures revealed that many of these unigenes reached their highest expression levels in R45min samples, especially antenna proteins (Appendix A), suggesting that the active recovery of chloroplasts and photosynthesis primarily started at this stage in the rehydration process of *R. canescens*, which is later than in some other desiccation-tolerant bryophytes [17,18]. This finding should be verified using direct physiological methods in the future.

In the R45min samples, more LEA protein and ELIP-coding genes showed the highest expression levels (Appendix A) and some unigenes related to GABA synthesis (Appendix A), suggesting that stress conditions may reappear at R45min. Photo-oxidation can occur if *T. ruralis* and some desiccation-tolerant lichens are rehydrated under light [65]. This may be the reason for the upregulation of unigenes encoding stress regulators. This was further supported by the increased activity of CAT, POD, and SOD detected in the R45min samples (Figure 2G–I), and many unigenes encoding enzymes for synthesizing flavonoids were also expressed at high levels simultaneously (Appendix A). Moreover, many unigenes encoding enzymes for carotenoid production showed the highest expression (Appendix A). Carotenoids may also be responsible for scavenging the singlet oxygen produced by photo-oxidation [66].

Our analysis of the R6h and R1d samples suggested that, at these time points, biological processes likely recovered to regular status, and R6h samples were more active in many aspects (Figure 7F). Most of the major stress-related genes were no longer expressed at high levels (Appendix A), and most physiological and biochemical indices remained high, except for CAT activity (Figure 2). Most unigenes related to energy metabolism and photosynthesis showed the highest expression levels in the R6h samples, and their expression was similar in the R1d samples (Appendix A). The unigenes related to the biosynthesis of sucrose, flavonoids, and phytohormones were also expressed at high levels in the R6h and R1d samples (Appendix A), suggesting that normal cellular activities may have nearly recovered, and cell division, plant growth, and differentiation may have resumed thereafter.

## 5. Conclusions

After rehydration, desiccated *R. canescens* can rapidly recover morphologically within minutes and physiologically within hours. Differentially accumulated proteins and genes differentially expressed in *R. canescens* at different rehydration stages were identified to explore the rapid recovery responses. The proteomics results revealed that, although the total protein content increased after rehydration, many types of proteins were likely degraded in the initial stages, indicating a rearrangement of the cellular functions at the translational level. The upregulated DAPs supported changes in the saccharide content, and the reconstruction of the cytoskeleton and chromatin during rehydration. The transcriptomic results suggest that the rehydration was not gradual, and that the vacuole appeared to play a significant role in the very early stages. Mitochondria and cell reproduction were detected to be potentially recovered before 30 min; photosynthesis maybe occurred at 45 min, and most biological functions likely resumed after 6 h. This study provides a foundation for further studies on desiccation-related processes in bryophytes. Overall, a large portion of DEGs and DAPs were already known in angiosperms, but were novel to bryophytes, and approximately 40% of the transcripts were not annotated. The precise functions of individual genes will be further investigated in the future. 

## Figures and Tables

**Figure 1 genes-14-00390-f001:**
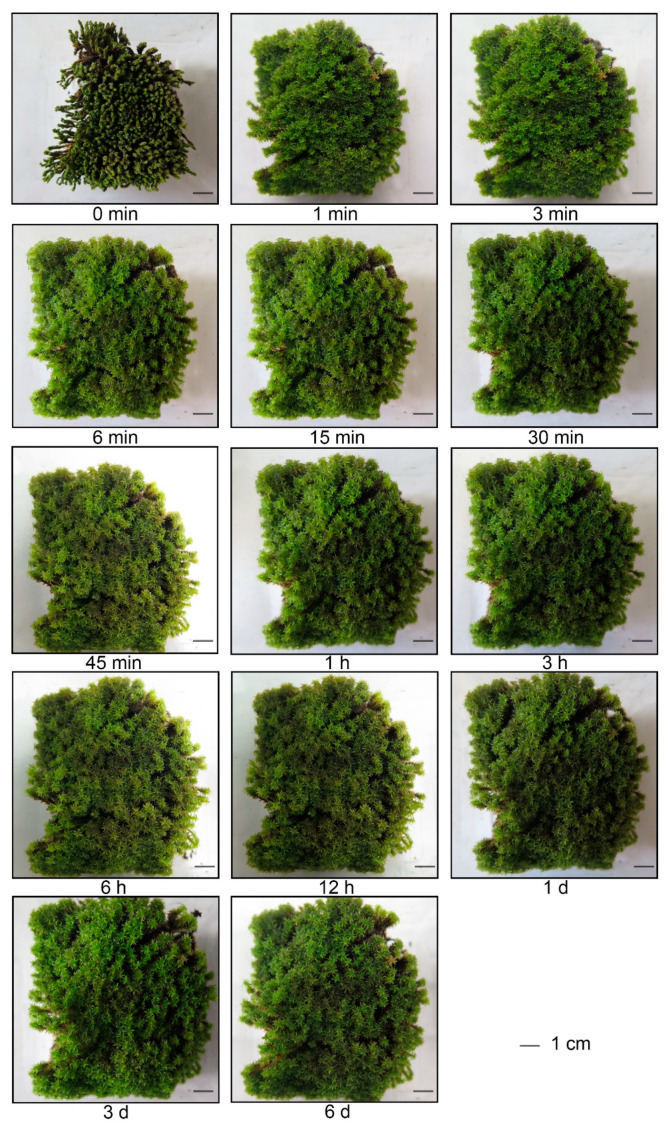
Morphological changes of desiccated *Racomitrium canescens* (*R. canescens*) in the rehydration processes.

**Figure 2 genes-14-00390-f002:**
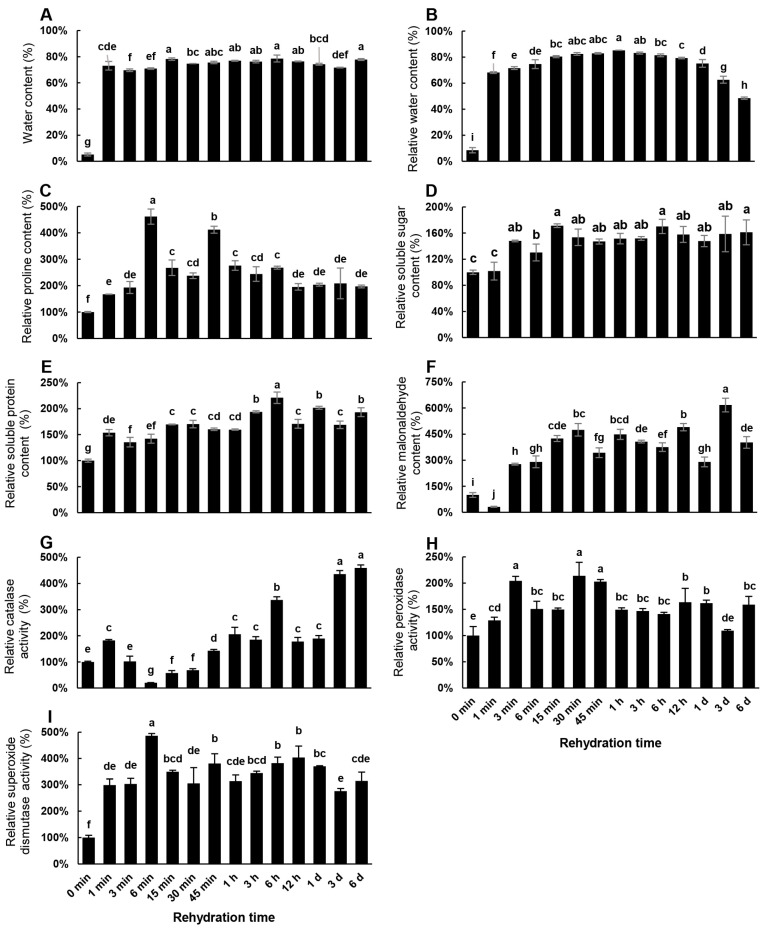
Physiological and biochemical changes of desiccated *R. canescens* in response to rehydration: (**A**) water content; (**B**) relative water content; (**C**) proline content; (**D**) soluble sugar contents; (**E**) soluble protein contents; (**F**) malonaldehyde content; (**G**) catalase activity; (**H**) peroxidase activity; (**I**) superoxide dismutase activity. Data are mean ± standard deviation (SD) of three replicates, letters above error bars indicate significance determined by one-way ANOVA using Duncan method by IBM SPSS 20.0 software (IBM, NY, USA), *p* < 0.05.

**Figure 3 genes-14-00390-f003:**
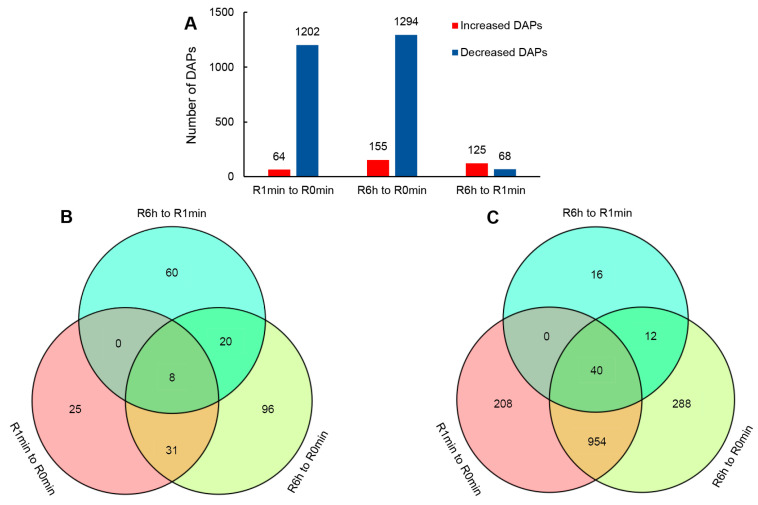
Summary of differentially accumulating proteins in *R. canescens* during the early stages of rehydration: (**A**) number of proteins whose abundance changed during at the first 1 min and 6 h of rehydration; (**B**) Venn diagram showing the extent of overlap between proteins that quickly accumulate in response to rehydration at various time points; (**C**) Venn diagram showing the extent of overlap between proteins whose abundance quickly decreases in response to rehydration at various time points. R0min, desiccated *R. canescens*; R1min, R6h: *R. canescens* rehydrated for 1 min or 6 h, respectively.

**Figure 4 genes-14-00390-f004:**
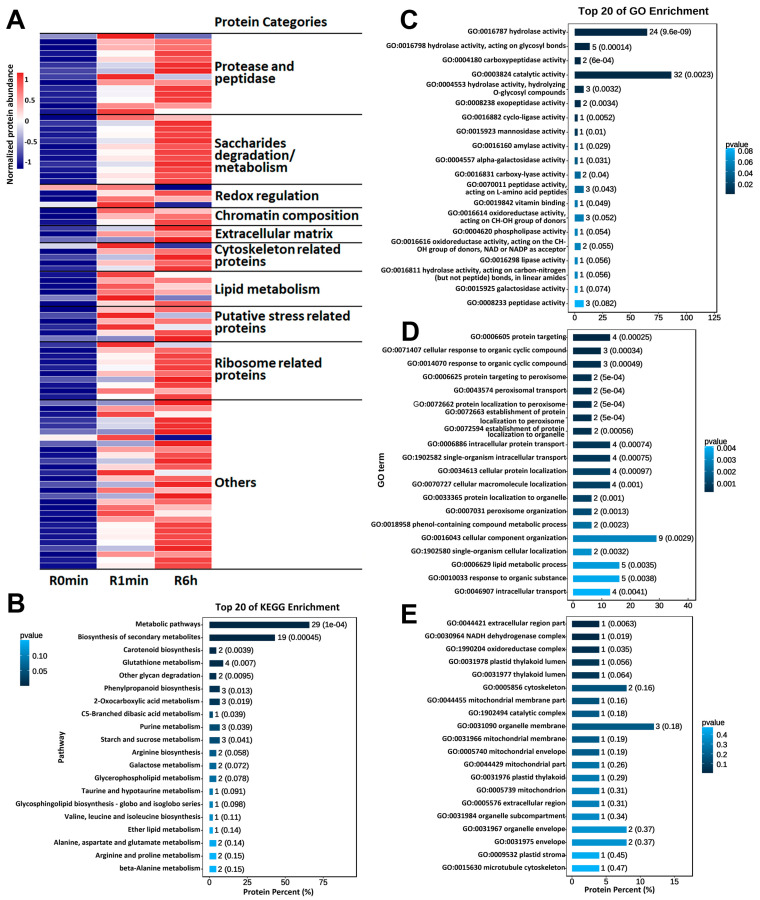
Relative amount, Kyoto Encyclopedia of Genes and Genomes (KEGG) database pathway enrichment, and Gene Ontology (GO) term enrichment of increased accumulated proteins: (**A**) heatmap representation of relative expression levels of increased accumulated proteins after rehydration; (**B**) the top 20 enriched KEGG pathways; (**C**) the top 20 enriched molecular function GO terms; (**D**) the top 20 enriched biological process GO terms; (**E**) the top 20 enriched cellular component GO terms. In heatmaps, the relative amount of each protein was normalized independently using the Z-score method. R0min, desiccated *R. canescens*; R1min, R6h: *R. canescens* rehydrated for 1 min or 6 h, respectively.

**Figure 5 genes-14-00390-f005:**
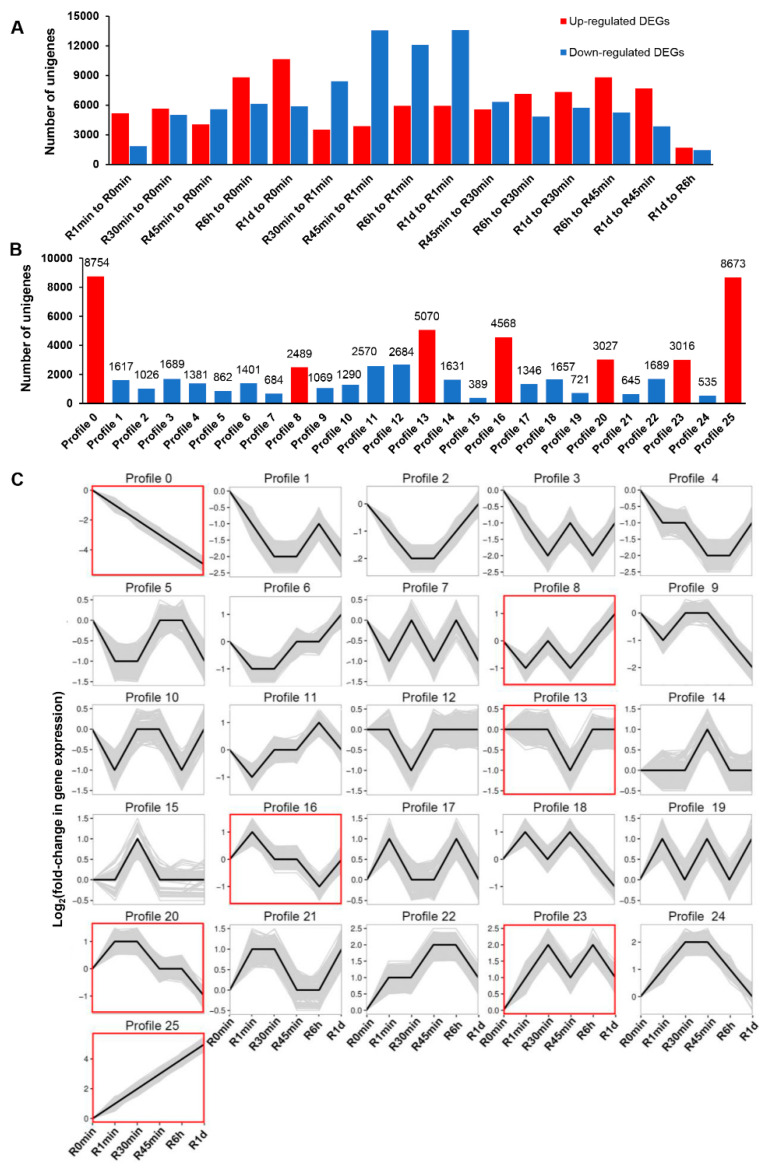
Summary of *R canescens* differentially expressed genes (DEGs) during the rehydration process: (**A**) number of DEGs between the indicated time points during rehydration; (**B**) number of DEGs associated with each cluster: red columns indicate significant clusters calculated by Short Time-Series Expression Miner software with a *Q*-value ≤ 0.05; (**C**) expression patterns of genes across all 26 clusters. Significant clusters have a red border.

**Figure 6 genes-14-00390-f006:**
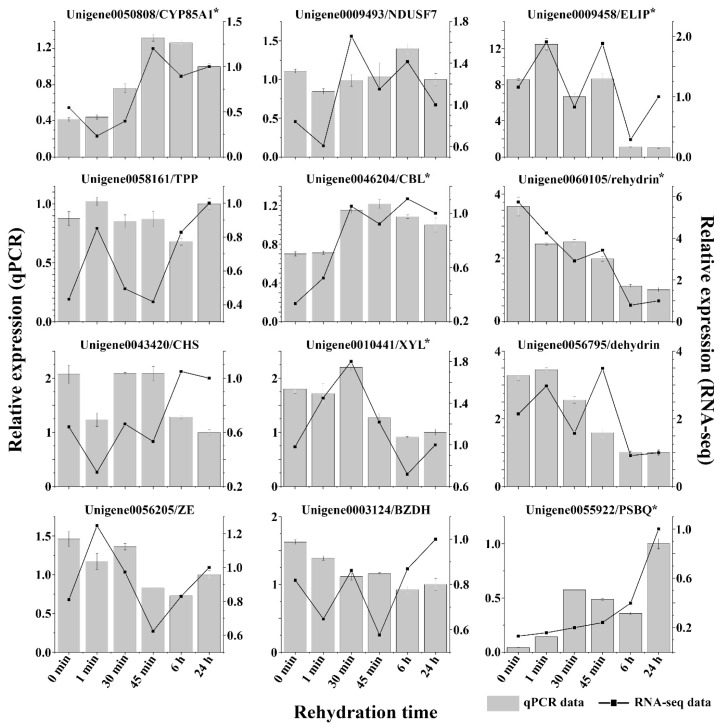
Validation of relative expression levels of unigenes from RNA-seq data using quantitative real-time PCR (qPCR). Relative expression levels of 12 DEGs were compared using RPKM values from RNA-seq results (linear graphs) and qPCR results (bar graphs), RPKM value and qPCR result of each unigene in the *R. canescens* samples rehydrated for 24 h were set to 1. Data are mean value of three replicates, error bars indicate standard deviation. Asterisks (*) indicate the Pearson correlation coefficient between qPCR and RNA-seq data is over 0.8. BZDH: benzaldehyde dehydrogenase, ELIP: early light-induced protein, NDUSF7: NADH dehydrogenase (ubiquinone) iron-sulfur protein 7, XYL: xylosidase, CHS: chalcone synthase, CBL: calcineurin B-like calcium binding protein, CYP85A1: cytochrome P450 85A1, PSBQ: photosystem II subunit QA, ZE: zeaxanthin epoxidase, TPP: trehalose-phosphatase.

**Figure 7 genes-14-00390-f007:**
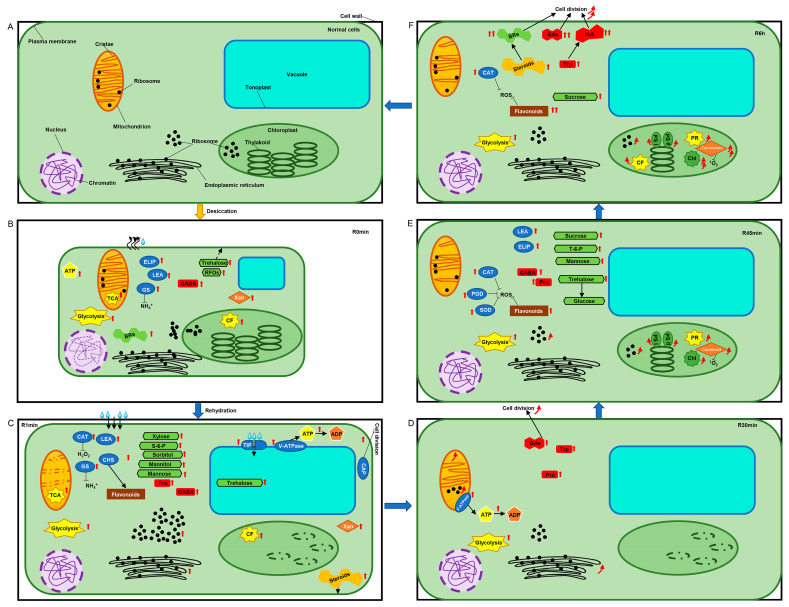
Proposed model of dehydration and rehydration processes in *R. canescens*. (**A**) normal condition before desiccation; (**B**) desiccated state; (**C**) 1 min of rehydration; (**D**) 30 min of rehydration; (**E**) 45 min of rehydration; (**F**) 6 h of rehydration. R0min: desiccated *R. canescens*; R1min, R30min, R45min, R6h: desiccated *R. canescens* rehydrated for 1 min, 30 min, 45 min, or 6 h, respectively. Red arrows indicate increased contents; curved red arrows indicate contents or structures were recovered. ^1^O_2_: singlet oxygen, ADP: adenosine diphosphate, ATP: adenosine triphosphate, BRs: brassinosteroids, CAP: cyclase-associated protein, CAT: catalase, CF: carbon fixation, Chl: chlorophyll, CHS: chalcone synthase, ELIP: early light-inducible protein, GABA: γ-aminobutyric acid, GAs: gibberellins, GS: glutamine synthase, H_2_O_2_: hydrogen peroxide, IAA: indoleacetic acid, LEA: late-embryogenesis abundant protein, POD: peroxidase, PR: photoreaction, Pro: proline, PS: photosystem, RFOs: raffinose family oligosaccharides, ROS: reactive oxygen species, S-6-P: sucrose-6-phosphate, SOD: superoxide dismutase, TCA: tricarboxylic acid cycle, TIP: tonoplast membrane intrinsic protein, Trp: tryptophan, Xan: xanthoxin.

## Data Availability

The mass spectrometry proteomics data were deposited in the Proteome Xchange Consortium via the PRIDE (http://www.ebi.ac.uk/pride, accessed on 9 January 2023) partner repository with the dataset identifier PXD026286. The RNA-seq dataset is available from the NCBI Short Read Archive (http://www.ncbi.nlm.nlm.nih.gow/sra, accessed on 9 January 2023) with accession number SRP319802.

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
