# Peer review of "Proteomic and Transcriptomic Responses of the Desiccation-Tolerant Moss Racomitrium canescens in the Rapid Rehydration Processes"

_genes, 2023, doi:10.3390/genes14020390_

Round 1

Reviewer 1 Report

This study leveraged multi-omics to analyzed proteomic and transcriptomic responses of the desiccation-tolerant Moss Racomitrium canescens, and propose a model to show the possible dehydration and rehydration processes, and the candidate genes would be interesting to validate in the other plants. But there are contents are missing in the main text and methods and should be added before further consideration.

On line 231. What tools used for the trimming reads?

On line 182. It would be better to describe how these activities calculated other than just cited another study. 

Figure 4. What does the legend bar mean? too small font for the enrichment tests.

On line 343, the main idea of results of the figure 4 should be described. Is that enrichment analysis corresponding to what you expected for the DAPs?

On line 345, ‘significance’ should provide pvalue.

On line 368 What’s the mapping ratio?

On line 381 how to select the 12 DEGs should be described.

On line 373 and line 379 what’s the KEGG and GO enrichment test results?

Reviewer 2 Report

Authors studied the physiological and biochemical changes of rehydration of desiccated R. canescens over 6 days. For more understanding on molecular pathways involved in desiccated R. canescens plants, they used proteomics and transcriptomics analysis. It highlighted different responsible pathways in desiccation tolerance of R. canescens plants. Biochemical changes were supported by proteomics and transcriptomics analysis. Results have been discussed clearly and supported by evidences. But there is couple of suggestion and question:

Figure 2: Use lowercase letter to label each column in all graphs.

Line 219: What is the selection criteria for fold changes of >1.2 and <0.83 (normally used >1.5 and <0.5 in proteomics study)?

Reviewer 3 Report

The MS is very well written and the topic is appropriate. There are only few mistakes:

Lines 34 drought without S

line 131 downcase canescens

line 132 comma before in this study

Line 170 clear point before JUST

line 184 downcase of FROZEN

line 214 ProteomeXchange

Line416:figure 6 in bold
